# Fabrication of Tapered Micropillars with High Aspect-Ratio Based on Deep X-ray Lithography

**DOI:** 10.3390/ma12132056

**Published:** 2019-06-26

**Authors:** Jae Man Park, Jong Hyun Kim, Jun Sae Han, Da Seul Shin, Sung Cheol Park, Seong Ho Son, Seong Jin Park

**Affiliations:** 1Department of Mechanical Engineering, Pohang University of Science and Technology (POSTECH), 77 Cheongam-Ro, Nam-Gu, Pohang, Gyeongbuk 37673, Korea; 2Pohang Accelerator Laboratory (PAL), Pohang University of Science and Technology (POSTECH), 77 Cheongam-Ro, Nam-Gu, Pohang, Gyeongbuk 37673, Korea; 3Department of Nano Manufacturing Technology, Korea Institute of Machinery & Materials (KIMM), 156, Gajeongbuk-ro, Yuseong-gu, Daejeon 34103, Korea; 4Surface Treatment R&D Group, Korea Institute of Industrial Technology (KITECH), 156 Gaetbeol-ro, Yeonsu-gu, Incheon 21999, Korea,

**Keywords:** tapered micropillars, high aspect-ratio, X-ray lithography, development

## Abstract

In this study, a fabrication method of tapered microstructures with high aspect ratio was proposed by deep X-ray lithography. Tapered microstructures with several hundred micrometers and high aspect ratio are demanded owing to the high applicability in the fields of various microelectromechanical systems (MEMS) such as optical components and microfluidic channels. However, as the pattern and gap size were downsized to smaller micro-scale with higher aspect ratio over 5, microstructures were easily deformed or clustered together due to capillary force during the drying process. Here, we describe a novel manufacturing process of tapered microstructures with high aspect ratio. To selectively block the deep X-ray irradiation, an X-ray mask was prepared via conventional ultraviolet (UV) lithography. A double X-ray exposure process with and without X-ray mask was applied to impose a two-step dose distribution on a photoresist. For the clear removal of the exposed region, the product was developed in the downward direction, which encourages a gravity-induced pulling force as well as a convective transport of the developer. After a drying process with the surface additive, tapered microstructures were successfully fabricated with a pattern size of 130 μm, gap size of 40 μm, and aspect ratio over 7.

## 1. Introduction

Synchrotron X-ray lithography has emerged as a novel micromachining technology to fabricate nano/micropillars with high aspect ratio and precise optics. The X-ray beam, which has high collimation and intensity, passes through a thick polymer and other materials and allows the materials to be shaped into a nearly vertical structure with good sidewall roughness and high pattern accuracy. With these advantages, X-ray lithography has been used to manufacture a sacrificial polymer mold [1,2], microfluidic channel [3], and tilted polymer structures [4].

As a further process, the nano/micropillars could be metal-electroformed as a mold insert and used for injection molding [5,6] and an embossing process [7]. This sequential process, called LIGA (German abbreviation: Lithographie, Galvanoformung, and Abformung) is beneficial for mass-production of micro-scale polymers and metal/ceramic structures with high aspect ratios. However, severe forces are exerted on the micropillars during the demolding process as the pattern and gap size become smaller in micro-scale and as the aspect ratio increases. As a result, the micropillars could be slanted or broken from the mold insert, and these defects deteriorate the quality of the final product. A mold insert with a tapered cavity could be a solution to facilitate the demolding process by reducing the interfacial area between the mold and micropillars. To fabricate a tapered mold insert, the inverted cylindrical microstructures that are pillar-shaped, known as a mandrel, also should be tapered before undergoing the electroforming process. It would be interesting to manufacture the tapered micropillars prior to electroforming because the conventional micromachining process does not realize a tapered shape.

To date, as a mandrel, research on the fabrication of tapered micropillars with high aspect ratio has examined the use of various exposure techniques. Klein et al. [8] fabricated 3 mm tapered tall posts using X-ray masks with low contrast during X-ray lithography. Using the same technique, Liu et al. [9] produced micro-scale pyramidal structures with aspect ratio under 5. Matsuzuka et al. [10] adopted a double X-ray exposure technique to attain a tapered microstructure, and tens of micro-scale tapered structures with aspect ratio under 3 were manufactured. Utsumi et al. [11] used an X-ray mask with distributed absorber thickness and replicated the trapezoidal cylindrical structures with aspect ratio below 3. Turner et al. [12] selected the multiple exposures of SU-8 and achieved a tapered SU-8 post with height over 1 mm. Most of them had an aspect ratio under 5, and the gap size was larger than the pattern size since critical slumping and clustering of micropillars could happen during the drying process.

Here, this study proposed and developed a novel technique using X-ray lithography to fabricate tapered micropillars with high aspect ratio. Double X-ray exposure with and without X-ray mask was conducted to impose a two-step dose distribution on photoresist [10]. The exposed region was placed in the downward direction to clearly eliminate a residual resist on the bottom of a substrate during the development process. In the drying process, surface additive was adopted to release a capillary force. Based on the optimized process, tapered micropillars with aspect ratio over 7 were successfully obtained without any visible defects.

## 2. Materials and Methods 

### 2.1. Experimental Setup

In this research, deep X-ray lithography experiments were performed at the 9D X-ray nano/micromachining beamline of the Pohang Accelerator Laboratory (PAL). Figure 1 represents the schematic diagram of the 9D beamline. A polychromatic X-ray beam was injected from a linear accelerator to a storage ring. The linearly accelerated electrons exhibited an energy of 3 GeV with beam current of 360 mA. As a core part in PAL, a bending magnet not only keeps the electrons in the storage ring, but it also delivers the X-ray source directly into the chamber [4]. The horizontal and vertical divergence of the X-ray source were 8 and 0.34 mrad, respectively. Before reaching the target sample in the chamber, the X-ray passes through a double mirror system, slit, and beryllium windows. The mirror system was used to cut off a high-energy-range X-ray, which could cause undesirable secondary effects such as backscattering and Auger electron effect [13]. By using a four-way slit aperture, the beam size on the sample can be adjusted to a square shape with a horizontal size of 100 mm and vertical length of 10 mm, which is sufficient for a 4-inch wafer. Beryllium windows were used for vacuum separation, and were surrounded by helium gas to prevent window oxidation. The chamber was also filled with helium gas to minimize the photon flux degradation and maintain a high X-ray transmission [14]. In addition, this gas gives an efficient convection of heat generation due to its non-zero heat capacity [4]. When the sample was mounted on the jig in the chamber, the sample stage reciprocated vertically to supply a uniform X-ray distribution on the photoresist. Table 1 summarizes the experimental setup condition of the 9D beamline.

### 2.2. Preparation of X-Ray Mask

During deep X-ray lithography, an X-ray mask was used to reduce the irradiated X-ray to a certain state, which cannot collapse a photoresist below the X-ray absorber. The X-ray mask consists of the X-ray absorber material and a substrate. Gold (Au) was adopted as an absorber material with the advantages of short attenuation length and easy process [15]. Polyimide (PI) film was employed as a flexible substrate with high transparency of X-ray and dimensional stability [16]. For the high-precision micropillars with high aspect ratio, it is essential to acquire a delicate X-ray absorber structure. In the present study, the X-ray mask was prepared by a conventional UV-lithography process with negative-type photoresist and Au electroplating process. Figure 2 displays the brief fabrication process of the X-ray mask. 

In the first step, the PI film was laminated on a polished 4-inch silicon wafer using an adhesive dry film resist (DFR). For the electroplating process, a seed layer composed of 20-nm Cr and 100-nm Au was deposited on PI film using an E-beam evaporation. The second step was to formulate a micropatterning on the seed layer. Prior to the patterning process, the Cr/Au seed layer was oxygen plasma-cleaned for 1 min. To create a uniform photoresist layer, a negative-type photoresist solution (SU-8 3010, MicroChem, Westborough, MA, USA) was spin-coated onto the seed layer at 600 rpm for 30 s, resulting in a thickness of about 25 μm. Then, the sample was lightly baked at 95 °C for 18 min. The photoresist layer was exposed to UV source of 10 mW for 17 s using a designed chrome mask, and subsequently post-exposure baking was carried out at 65 °C for 1 min and at 95 °C for 4 min. To dissolve the unexposed region, the specimen was immersed in SU-8 developer for 5 min, followed by rinsing with isopropyl alcohol for 1 min. Finally, residual photoresist was eliminated via oxygen plasma cleaning for 15 s. The third step was to electroplate the Au layer on the developed SU-8 area. An electroplating process was performed at 50 °C using a customized electrochemical cell, which is filled with deionized water and electrolyte gold solution. The developed seed layer region was Au electroplated with current density of 1 mA/cm^2^ for 7 h. Thereafter, the substrate was sufficiently rinsed by distilled water. In the final stage, the X-ray mask was detached from the silicon wafer. Table 2 illustrates the processing conditions of X-ray mask fabrication used in this study.

### 2.3. X-Ray Exposure

As a positive-type photoresist, a poly(methyl methacrylate) (PMMA; Goodfellow) with a thickness of 980 μm was used for X-ray irradiation by attaching it to a base substrate. The PMMA is generally used for deep X-ray lithography due to its freestanding contrast and process stability. Prior to adhering it on a substrate, PMMA was polished and annealed at 80 °C for 1 h to relax its internal stress. As a base substrate, a polished graphite sheet with thickness of 2 mm was prepared because of its high porosity, micro-roughness, and low secondary radiation effect [17]. These properties are advantages for the excellent adhesion with PMMA. The graphite was spin-coated with 1000 rpm using a liquid PMMA (950 PMMA C9; MicroChem, Westborough, MA, USA) and baked at 180 °C for 1 h. This process was repeated four times to ensure a sufficient PMMA adhesion layer. Thereafter, the PMMA sheet was chemically bonded to the graphite sheet using a pre-developed solution [18]. An X-ray exposure process was performed on the substrate with the prepared X-ray mask via a scanning system of the 9D beamline. The X-ray mask was placed on the top surface of PMMA in the jig, and the mask was covered by PI film. For the reduction of X-ray diffraction in air atmosphere, the substrate, X-ray mask, and PI film were hard contacted by sucking out the air using a vacuum system. Then, the chamber containing the jig and reciprocating part was filled with helium before X-ray exposure. In this study, double X-ray exposure process with and without X-ray mask was performed to give a two-step dose distribution on the PMMA. First, the PMMA with X-ray mask was exposed to the irradiated X-ray with exposure energy of 4.5 kJ/cm^3^. Second, both the exposed and unexposed PMMA regions without X-ray mask were X-ray irradiated once more with exposure energy of 0.25 kJ/cm^3^. The detailed specification of double X-ray exposure is given for Table 3.

### 2.4. Development

To etch out the exposed region, a GG-developer consisting of 60 vol.% 2-(2-butoxyethoxy) ethanol, 20 vol.% tetrahydro-1, 4-oxazine, 5 vol.% 2-aminoethanol, and 15 vol.% deionized water was used for 6 h. After the development process, residual PMMA on the substrate was dissolved by GG rinsing solution including 80 vol.% 2-(2-butoxyethoxy) ethanol and 20 vol.% deionized water. Finally, the sample was finished after the cleaning process using a deionized (DI) water mixed with surface additive (BYK-3440; Uni Trading Corporation, Gwangju, Korea).

## 3. Results and Discussion

### 3.1. Interaction between X-Ray Source and Photoresist

Au X-ray mask on the top of the PMMA should be able to efficiently block the deep X-ray as an absorber material. Figure 3a displays the schematic process of X-ray lithography. The X-ray can easily penetrate the SU-8 micropattern on the PI film because the material is transparent to X-rays. On the other hand, the gold layer showed low X-ray transmittance due to the high absorption property of gold to X-ray, as shown in Figure 3b. With the increase of Au thickness, it became more difficult for X-ray to transmit through the Au layer. Before the X-ray reached the X-ray absorber, the photon flux distribution of the light source was analyzed, since it greatly influences the quality of the microstructure. The photon flux distribution was calculated by reference to inherent properties of the beam source. From the analysis, we can predict the desirable Au thickness, and interaction between the electrons and photoresist materials. The photon flux distribution generated from the X-ray at 3.0 GeV showed a wide range of energies from a few volts to tens of kilo-electron volts, as indicated in Figure 3c. When the X-ray passed through two Be windows, its photon flux was reduced to 66% of the original beam source while still maintaining high-energy photons. These high-energy photons were absorbed in the substrate of the photoresist, resulting in creation of photoelectrons, Auger-electrons, and fluorescence photons. Such photons were backscattered to the PMMA layer, which reduced the molecular weight of PMMA on the interfacial surface [19]. Consequently, it caused a poor adhesion between PMMA and the substrate. To eliminate high-energy photons with energies over 7 keV, a double mirror system with angle of 0.45° was used and revealed a flux distribution with low-energy photon range.

With the prepared beam source, the thickness of the X-ray mask was calculated using the X-ray flux and dose incident on the PMMA. The number of transmitted X-ray spectrum ($s_{TP}$) in the perpendicular cross-section of the PMMA can be defined by the following equation [20].
(1)$$s_{TP}=S_{M}\times \exp\left(-\mu_{p}x_{p}\right)$$
where $S_{M}$ is the X-ray energy spectrum of the photoresist, $\mu_{p}$ is the absorption coefficient of photoresist, and $x_{p}$ is the optional depth. Based on the calculation, the dose of the absorber top should be less than 100 J/cm^3^ to avoid the development. Considering the requirement, the Au thickness of the X-ray mask must be over 8 μm. In this study, the final thickness of the X-ray mask during the electroplating process was set to 21 μm, which is a thick enough layer to prevent the development of the absorber top and minimize the back-scattering effect. Figure 4 reveals the prepared X-ray mask with diameter of 130 μm and gap size of 40 μm.

### 3.2. Double X-Ray Exposure

In this research, a double X-ray exposure process with and without an X-ray mask was adopted to acquire tapered micropillars with high aspect ratio. This method gives a two-step dose distribution in the PMMA and results in a tapered shape during the development process [10]. If a single X-ray exposure with X-ray mask were performed on the PMMA, micropillars would have a vertical shape without any taper. In the first step, PMMA adhered to an X-ray mask was irradiated with exposure energy of 4.5 kJ/cm^3^, which is enough energy to eliminate the exposed PMMA region. During X-ray exposure, a dose on the exposed top surface over 20 kJ/cm^3^ might cause a thermal swelling of the PMMA. Based on the reference to inherent characters of beam source, the amount of dose on the exposed top surface was calculated and showed a value of 19.98 kJ/cm^3^. As a result, the exposed region of the PMMA showed a slight amount of thermal swelling because the amount of dose nearly reached the theoretical limit, as shown in Figure 5a. In the second exposure, both exposed and unexposed regions during the first exposure were X-ray irradiated due to the absence of an X-ray mask. The excessive amount of exposure dose not only induced a thermal swelling for the exposed region, but it also drastically lowered the aspect ratio of micropillars for the unexposed region. With the consideration of all requirements, the exposure dose in the second exposure was determined to be 0.25 kJ/cm^3^. Figure 5b represents the result of double X-ray exposure. The double exposed region having a top dose of 20.02 kJ/cm^3^ showed more thermal swelling since the dose on the top surface exceeded 20 kJ/cm^3^.

### 3.3. Stability of Micro Pillar Structure

To obtain the final tapered micropillars, the exposed area of PMMA was developed using a GG-developer. The product was mounted in the downward direction to efficiently remove the exposed region. This method uses a gravity-induced pulling force as well as convective transport of the GG-developer and results in more thorough removal of PMMA and reduction of development time [21]. After GG-rinsing, the sample was cleaned by pure deionized (DI) water to dilute a residual GG-solution. However, the micropillars showed slight fracture on the tip and were clustered together due to capillary force when DI water was evaporated off the surface, as shown in Figure 6. To improve these defects by capillary force, DI water was mixed with surface additives, and the resulting contact angle was compared, as displayed in Figure 7.

The mixed solution resulted in a smaller contact angle, meaning the solution had a weaker capillary force during the evaporation process [22]. Taking advantage of this effect, tapered micropillars were successfully fabricated with pattern size of 130 μm, gap size of 40 μm, and aspect ratio over 7, and showed no fractures or clustering, as revealed in Figure 8.

## 4. Conclusions

In this study, a novel fabrication technique of tapered micropillars with high aspect ratio was proposed and demonstrated by X-ray lithography. An X-ray mask with thickness of 21 μm was prepared to efficiently block deep X-rays and minimize back-scattering effect. Double X-ray exposure with and without X-ray mask was conducted by imposing a two-step dose distribution on the PMMA. After the exposure, the top surface of the double-exposed region showed slight thermal swelling because the value exceeded the theoretical limit of 20 kJ/cm^3^. Thereafter, the exposed region was developed in the downward direction to clearly eliminate the PMMA residual on the bottom surface. To release a capillary force during the drying process, DI water was mixed with surface additive to achieve a smaller contact angle than pure DI water. Taking advantage of the release effect, tapered micropillars with pattern size of 130 μm, gap size of 40 μm, and aspect ratio over 7 did not show any clustering phenomenon or fractures on the tips. Compared with no tapered shape, tapered micropillars would provide an easy demolding process by reducing the surface area between the electroformed mold insert and metal/ceramic micro structures during manufacturing process. In addition, it is expected that this novel method could be used for manufacturing tapered micropillars with high aspect ratio in the MEMS field, including optical components and microfluidic channels.

## Figures and Tables

**Figure 1 materials-12-02056-f001:**
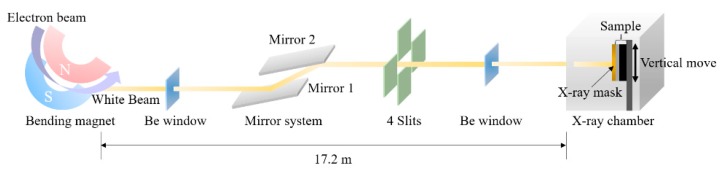
Schematic diagram of 9D beamline.

**Figure 2 materials-12-02056-f002:**
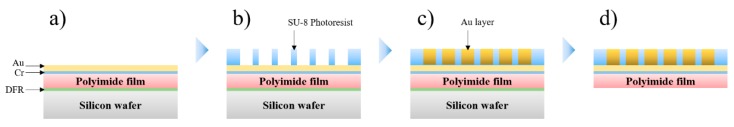
Fabrication process of X-ray mask: (**a**) deposition of Cr and Au seed layer on polyimide film; (**b**) micropatterning process on the seed layer by UV lithography; (**c**) Au electroplating between the mask patterns; and (**d**) X-ray mask detached from the silicon wafer.

**Figure 3 materials-12-02056-f003:**
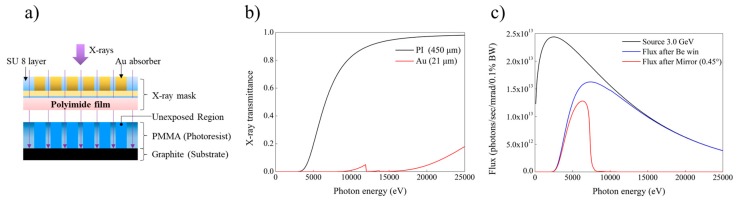
Experiment conditions of deep X-ray lithography: (**a**) schematic diagram of X-ray irradiation into photoresist via X-ray mask; (**b**) X-ray transmittance with respect to different materials; and (**c**) calculation of X-ray flux before entering the lithography chamber.

**Figure 4 materials-12-02056-f004:**
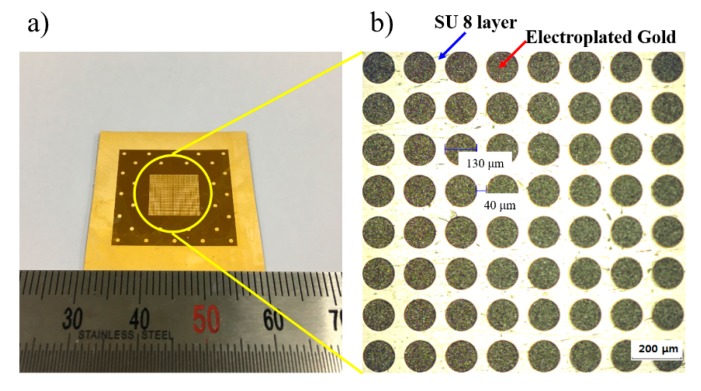
Fabricated X-ray mask for X-ray exposure: (**a**) entire image of X-ray mask prepared by UV lithography; and (**b**) magnified feature of X-ray mask having circle shape.

**Figure 5 materials-12-02056-f005:**
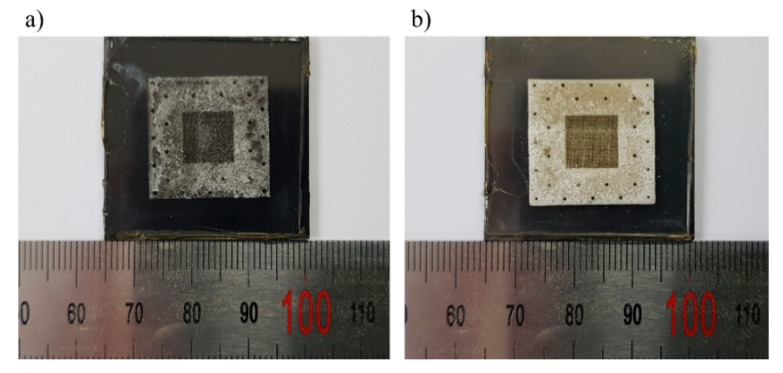
The results of X-ray exposure: (**a**) first exposure with X-ray mask showing a little thermal swelling; and (**b**) second exposure without X-ray mask indicating a more thermal swelling.

**Figure 6 materials-12-02056-f006:**
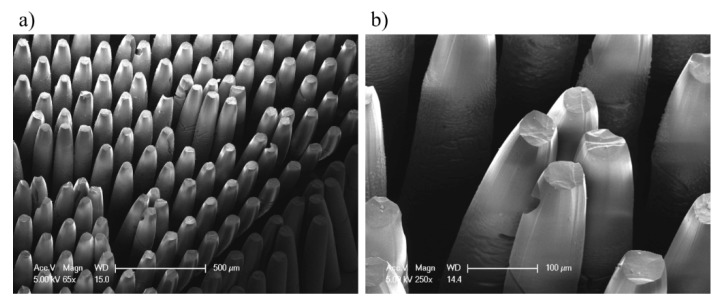
The results of drying process: (**a**) clustered tapered micropillars each other due to strong capillary force; and (**b**) magnified image of clustered and fractured micropillars.

**Figure 7 materials-12-02056-f007:**
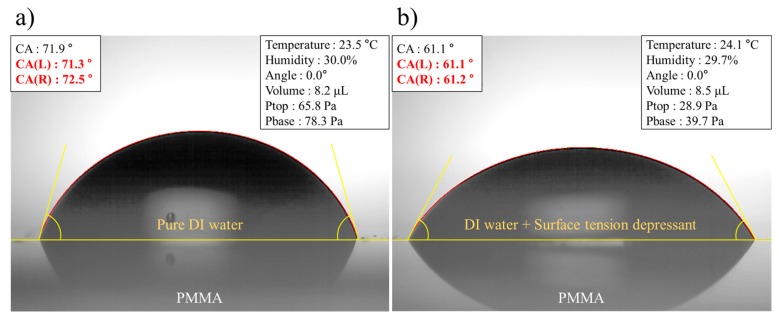
The measurement of contact angle: (**a**) pure DI water; and (**b**) DI water mixed with surface addictive.

**Figure 8 materials-12-02056-f008:**
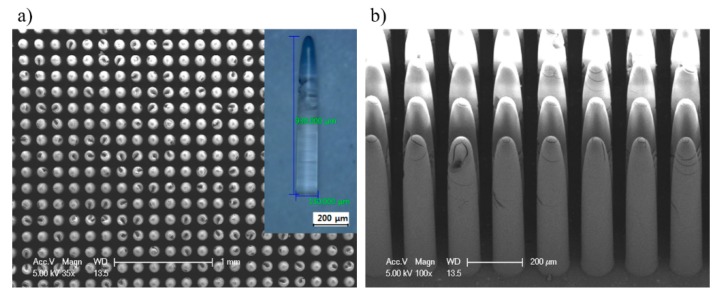
Fabricated tapered micropillars with high aspect-ratio: (**a**) top view; and (**b**) side view.

**Table 1 materials-12-02056-t001:** Experimental setup condition of 9D beamline.

Setup Condition	9D Beamline
Original photon energy	3.0 GeV
Mirrored photon energy	3.0 to 7.0 KeV
Beam current	360 mA
Horizontal beam divergence	8 mrad
Vertical beam divergence	0.34 mrad
Beam size	100 mm (H) × 10 mm (V)
Sample environment	Vacuum or He atmosphere

**Table 2 materials-12-02056-t002:** Processing conditions of X-ray mask fabrication.

**Preparation of Substrate**	
Base Substrate	4 inch silicon wafer
Seed layer	Cr: 20 nm/Au: 100 nm
**UV lithography Process**	
Surface cleaning by oxygen plasma	300 W for 1 min
Spin coating	600 rpm for 30 s (thickness: 25 μm)
Soft baking	95 °C for 18 min
UV exposure energy and time	170 W / 17 s
Post exposure baking	65 °C for 1 min / 95 °C for 4 min
Development	5 min
Surface cleaning by oxygen plasma	300 W for 15 s
**Au electroplating Process**	
Applied current	64 mA for 7 h
Current density	1 mA/cm^2^

**Table 3 materials-12-02056-t003:** The detailed specification of double X-ray exposure.

Controllable Variables	1st Exposure with X-ray Mask	2nd Exposure without X-ray Mask
Mirror angle (°)	0.45	0.40
Exposure energy (kJ/cm^3^)	4.5	0.25
Filter thickness (μm)	18	0

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
