# Peer review of "Fabrication of Tapered Micropillars with High Aspect-Ratio Based on Deep X-ray Lithography"

_materials, 2019, doi:10.3390/ma12132056_

Reviewer 1 Report

The paper by Park is the example of the study describing the application of hybrid X-ray and UV lithography in microelectronics. This paper is well-written, so it can be published in present form.

Author Response

Dear reviewer 1,

Thank you for your comments about the paper.

Considering all reviews including other reviewers, the manuscript was newly revised.

The revised part was yellow-highlighted in the manuscript.

Sincerely,

Seong Jin Park.

Reviewer 2 Report

The manuscript by J.M. Park et al. shows an elegant solution to the problem of capillary forces for microstructure manufacturing. While microstructures of similar dimensions typically collapse a lower aspect ratios, the authors introduce a surfactant to extend the reachable thickness.

While this work is technically and scientifically sound, it addresses a very specific problem in a specific field. Due to the focus on this particular issue, the paper reminds in some parts of a lab report. In specific, I miss the comparison to other techniques and a general classification of the method in the discussion part. I would appreciate, if the authors could introduce some sentences why other techiques cannot be easily applied to their specific problem.

Coming from nanofabrication of much smaller structures, I was initially surprised why an aspect ratio of 7 is such an achievement. In fabrication of micro-/nanostructures in silicon, for example, aspect ratios of 40 and more can be achieved when capillary forces are avoided (see, for example Lebugle et al, doi: https://doi.org/10.1364/AO.57.009032). The point of the authors would be much stronger, if they could introduce two to three sentences that compare their technology to other methods, such as critical point drying as used in the previously mentioned study. It became just clear to me after some thinking that PMMA in this size is much more difficult to handle, and that method such as critical point drying cannot be applied to it. A helpful review I found could also be del Campo et al., Chem. Rev. 2008, 108, p. 911-945 (doi: https://doi.org/10.1021/cr050018y).

All in all, I recommend publication with minor revisions, if the following comments are addressed appropriately:

Title: I suggest to change "Microstructures" to "Micropillars", to be more specific. Microstructures is a bit too general in light of the presented results.

Line 34: Introduce "beam" after "X-ray"

Line 48: "Reverse microstructures" (again a very general term) are described as "mandrel", which is a very specific form. This should be stated more explicitely. One suggestion could be "inverted cylindrical microstructures that are pillar-shaped, also known as mandrel"

Same paragraph: Some information why the fabricaiton of mandrels is interesting would be nice.

Line 50: "tapered microstructures": why not "micropillars" or the newly introduced term "mandrel"?

Line 79: Capitalize "Auger"

Line 85: Helium is an ideal gas. Ideal gases have, as a matter of fact in thermodynamics, the lowest heat capacity of all gases (the reason is that mono-atomic gases cannot rotate neither vibrate, and gas molecules that can rotate and vibrate can take up more energy). I suggest to leave out the phrase "due to its high heat capacity". The intention of the sentence remains unchanged by that as it is obvious that gas can transport heat, and vacuum not. If you would like to keep the part of the sentence, "high" should at least be changed to "non-zero".

Line 91, Table 1: At this stage, I missed the information on the photon energy. I saw that it is described later in detail. Maybe you could just add a line with "photon energy    approx. 3-7 keV"?

Line 94: The context of "collapse" was unclear to me. Do you mean "expose (sufficiently)"?

Line 123: Again I missed something that was described later: A number of resulting Au thickness should be introduced (without any details, the description on page 5 is sufficient).

Conclusions: The paragraph is more a summary of the work than conclusions. Only the last sentence deals with it. Could you introduce 2-3 sentences why the tapered micropillars are important in the field of MEMS? Is there another interesting application?

Author Response

Dear reviewer 2,

Thank you for your comments about the paper.

Considering all reviews including other reviewers, the manuscript was newly revised.

The revised part was yellow-highlighted in the manuscript.

Sincerely,

Seong Jin Park.

========================================================================

Line 94: The context of "collapse" was unclear to me. Do you mean "expose (sufficiently)"?

X-ray mask was used to selectively block the deep X-ray beam. The exposed region would be dissolved during the development process. On the other hand, the unexposed region would not be dissolved during the development process if the gold on the X-ray mask efficiently blocks the X-ray. In that sentence, it would be great to understand the word "collapse" to a similar meaning "dissolution during the development process".

Reviewer 3 Report

The paper claims the highest aspect ratio of the literature, which is not completely exact.

Although in the literature some higher aspects ratio have been obtained in PMMA, the combination of high aspect ratio + high density of no sticking or collapsing structures is new.

The paper is clearly written and the process to obtain such a feature is clearly described.

The paper can be published as is.

Reviewer 4 Report

This is a review of the paper titled: “Fabrication of Tapered Microstructures with High 3 Aspect-Ratio based on Deep X-ray Lithography”.

In this paper, authors are describing a new fabrication method for random micro-structuring the surface. They used an x-ray mask to pattern the surface using x-ray lithography. Their claim is to overcome collapsing of high aspect ratio microstructures due to capillary forces over rinsing and drying process.

I recommend a minor revision before publication. Comments are as follow:

_ It is not clear why the authors were used double X-ray exposure process with and without an X-ray mask. As a reference sample, a single X-ray exposure should be processed, and the results should be discussed.

Author Response

Dear reviewer 3,

Thank you for your comments about the paper.

Considering all reviews including other reviewers, the manuscript was newly revised.

The revised part was yellow-highlighted in the manuscript.

Sincerely,

Seong Jin Park.